# Role of Bioactive Compounds in the Regulation of Mitochondrial Dysfunctions in Brain and Age-Related Neurodegenerative Diseases

**DOI:** 10.3390/cells11020257

**Published:** 2022-01-13

**Authors:** Khadidja Kessas, Zhor Chouari, Imen Ghzaiel, Amira Zarrouk, Mohamed Ksila, Taoufik Ghrairi, Adil El Midaoui, Gérard Lizard, Omar Kharoubi

**Affiliations:** 1Laboratory of Biotoxicology Experimentale, Biodepollution and Phytoremediation, Faculty of Life and Natural Sciences, University Oran1 ABB, Oran 31100, Algeria; khadoujakess@gmail.com (K.K.); chouarizhor33@gmail.com (Z.C.); 2Team ‘Biochemistry of the Peroxisome, Inflammation and Lipid Metabolism’ EA7270/Inserm, University Bourgogne Franche-Comté, 21000 Dijon, France; imenghzaiel93@gmail.com (I.G.); mohamed.ksila@fst.utm.tn (M.K.); gerard.lizard@u-bourgogne.fr (G.L.); 3Faculty of Medicine, LR12ES05, Lab-NAFS ‘Nutrition—Functional Food & Vascular Health’, University of Monastir, Monastir 5000, Tunisia; zarroukamira@gmail.com; 4Faculty of Sciences of Tunis, University Tunis-El Manar, Tunis 2092, Tunisia; 5Faculty of Medicine, University of Sousse, Sousse 4000, Tunisia; 6Laboratory of Neurophysiology, Cellular Physiopathology and Valorisation of BioMoleecules, LR18ES03, Department of Biologie, Faculty of Sciences, University Tunis-El Manar, Tunis 2092, Tunisia; taoufik.ghrairi@fst.utm.tn; 7Research Team “Biology: Environment and Health”, Department of Biology, Faculty of Sciences and Techniques Errachidia, Moulay Ismail University of Meknes, Errachidia 52000, Morocco; adil.el.midaoui@umontreal.ca; 8Department of Pharmacology and Physiology, Faculty of Medicine, Université de Montréal, Montréal, QC H3C 3J7, Canada

**Keywords:** antioxidant therapy, ageing, energy metabolism, mitochondria, neurodegenerative diseases

## Abstract

Mitochondria are multifunctional organelles that participate in a wide range of metabolic processes, including energy production and biomolecule synthesis. The morphology and distribution of intracellular mitochondria change dynamically, reflecting a cell’s metabolic activity. Oxidative stress is defined as a mismatch between the body’s ability to neutralise and eliminate reactive oxygen and nitrogen species (ROS and RNS). A determination of mitochondria failure in increasing oxidative stress, as well as its implications in neurodegenerative illnesses and apoptosis, is a significant developmental process of focus in this review. The neuroprotective effects of bioactive compounds linked to neuronal regulation, as well as related neuronal development abnormalities, will be investigated. In conclusion, the study of secondary components and the use of mitochondrial features in the analysis of various neurodevelopmental diseases has enabled the development of a new class of mitochondrial-targeted pharmaceuticals capable of alleviating neurodegenerative disease states and enabling longevity and healthy ageing for the vast majority of people.

## 1. Introduction

During rapid proliferation, the molecular mechanisms involved in cell growth require mitochondrial energy products made by aerobic glycolysis [1]. Several cell types depend on mitochondria for energy synthesis to varying degrees. Because mitochondria are a cell’s main powerhouse, the use of uncouplers of respiration and oxidative phosphorylation at the phase of AMP-dependent protein kinase activation could also prevent negative consequences of the cellular hyper-energisation [2]. The nervous system has neuroplasticity characteristics, which are a range of adaptive changes that help in response to physiological or pathological perturbations. The biological basis for structural and functional adaptability potential is provided by a variety of cellular and molecular pathways, for example, by glutamate, which is a basic excitatory neurotransmitter [3,4]. Brain-derived neurotrophic factor [5] and neural cell adhesion molecule [6] are the two best researched of these signals. Cell signalling pathways are essential for function regulation and energy production. However, researchers address the relationship of the latter signalling pathways with mitochondria, an organelle that, in addition to being a power station, is increasingly becoming recognised as a signalling platform involved in basic processes in the development and plasticity of brain circuits. In addition to converting energy substrates into ATP, mitochondria are involved in the metabolism of reactive oxygen species (ROS), calcium homeostasis, and apoptosis [7]. Mitochondria can be located along axons and are actively attracted to presynapses. Mitochondria are mainly found in dendritic shafts; however, they can also be found near spines [8]. The application of novel imaging and molecular biology technologies to mitochondrial studies has revealed several surprising properties and functions of mitochondria in neuroplasticity, including (-) rapid movement within and between subcellular compartments [9]; (-) fission and fusion, which have been facilitated by two distinct protein complexes incorporating GTPase [10]; (-) response to electrical activity and stimulation of neurotransmitter and growth factor receptors [11]; (-) serve as signalling outposts for kinases, deacetylases, and other signal transduction enzymes [12].

Recognising the centrality of oxidative stress in neuronal disorders, there is a growing interest in the development of antioxidant therapeutics that could potentially replicate the physiological processes of the natural antioxidant defensive system, which is primarily comprised of enzymatic components such as superoxide dismutase, catalase, and glutathione peroxidase [13,14,15]. More specifically, there has been significant research focus on using antioxidants for neuroprotection, which is a therapy alternative for central nervous system illnesses that target oxidative stress and excitotoxicity [16]. Antioxidants are natural or unnatural chemical compounds that, at low concentrations, potentially reduce or eliminate the consequences of oxidative stress by combating the damaging consequences of ROS/RNS [17]. Antioxidants have anti-inflammatory, anti-bacterial, anti-cancer, cardioprotective, and neuroprotective capabilities, and there is evidence supporting their potential therapeutic effects on diabetes, arthritis, and osteoporosis [13,18]. However, the antioxidant medicines approved for human use in central nervous system disorders remain relatively low, suggesting that further extensive work in the domain of antioxidant medication development is required [19,20]. Furthermore, the absence and/or presence of ROS/RNS imbalances limits the identification of relevant antioxidant components with beneficial effects. In this context, the contribution of this project is to present an overview of oxidative stress processes and antioxidant therapy for neuroprotection to make them more accessible.

## 2. Involvement of Mitochondria in Neuronal Functions

Mitochondria are extremely important in the brain because they provide the energy for neurons to function properly, especially at synapses, which have a high energy requirement [21]. Mitochondria are intracellular organelles present in numerous copies in all nucleated cells, each with its own genome [22]. This organelle is recognised as the powerhouse of the brain because of its high energy demand [23,24]. Disruption of mitochondrial ROS generation, particularly in the CNS, has been associated to the development of ischemia-reperfusion damage and chronic neurodegenerative disorders (e.g., Alzheimer’s and Parkinson’s disease) [25]. Another important consequence of Ca^2+^ transport and mitochondrial ROS production in pathophysiological conditions is the interaction between mitochondrial function and neuronal activity [26,27]. However, under normal conditions, it generates mild increases in ROS production, influencing intracellular signalling cascades and endogenous glutamate release [28], synaptic transmission [29], and communication between neurons and glial cells. Furthermore, mitochondria are essential for many aspects of neurodevelopment and neuronal function, as well as important regulators of cellular activities such as bioenergetics, calcium homeostasis, redox signalling, and apoptotic cell death [30]. In the mitochondrial respiratory chain, Complex IV (cytochrome oxidase) retains all partially reduced intermediates until full reduction is achieved. Other redox centres in the electron transport chain, on the other hand, may leak electrons to oxygen, partly reducing it to a superoxide anion (O_2_^•−^) [31]. In mammalian mitochondria, a total of 11 sites that generate superoxide (O_2_) and/or hydrogen peroxide (H_2_O_2_) and are related to substrate oxidation have been identified [32,33]. It has been proposed that mitochondria may possibly play a role in certain types of synaptic plasticity. After high-frequency stimulation, synaptic strength increases, resulting in post-tetanic potentiation. The fundamental mechanism is a short-term increase in synaptic release probability caused by residual Ca^2+^ in the presynaptic terminal during intense stimulation [34].

The generation of superoxide radicals (O_2_^•−^) might affect the mitochondrial membrane potential (Δψm). Mitochondria can regulate glycolysis, the activity of Ca^2+^ and Na^+^-K^+^-ATPases at the plasma membrane, and hence, the activity of Na^+^-coupled plasma membrane transporters via the ATP/ADP pool. Neuronal growth, neurotransmitter release, and plasticity are all influenced by changes in cytosolic free calcium ion concentration [Ca^2+^]c. Mitochondrial Ca^2+^ uptake alters the spatiotemporal pattern of cytosolic Ca^2+^ signals in the physiological range and may consequently influence neuronal activity. The sluggish recycling of Ca^2+^ into the cytosol by mitochondrial Ca^2+^ buffering reduces the peak amplitude of stimulus-induced [Ca^2+^]c transients and slows [Ca^2+^]c recovery [35]. Aside from energy generation, mitochondria govern crucial components in the activation of cell signalling pathways, such as survival, proliferation, and differentiation, as well as the maintenance of ionic gradients in cell membranes [36]. Neurogenesis, or the generation of new neurons from stem cells, happens rapidly throughout nervous system development and to a much lesser amount in some adult nervous system regions. Adult neurogenesis is thought to be functionally significant as a process of brain plasticity in pathological situations and in brain healing following injury [37,38]. Although it is widely understood that neurogenesis is the differentiation of neural progenitors into post-mitotic neurons, some research has been conducted to learn more about the molecular alterations that occur during this neuronal differentiation [39]. It has been determined that mitochondrial translation products might be involved in the structure of the cytoskeleton or of certain membrane components, whose rearrangements should be prerequisites or correlates to early phases of neural development. Increased ATP synthesis in mitochondria is a mechanism involving mitochondrial activation in neurite development. Moreover, the mitochondrial uncoupling of proteins has gotten a great deal of interest for their functions in physiological processes other than heat generation in brown fat cells [40]. Uncoupling proteins (UCPs) diminish ATP production, reduce ROS formation, and alter mitochondrial and endoplasmic reticulum Ca^2+^ dynamics by leaking protons across the mitochondrial inner membrane [41]. Neurogenesis is influenced by a variety of molecular pathways. Transcriptional gene regulation is an example of a process that governs neurogenesis [42]. Many steps of neurogenesis are also influenced by extrinsic factors such as signalling molecules [43]. Recently, research into the processes of neurogenesis has broadened to include lipid metabolism [44]. Fatty acid oxidation is essential for Niemann-Pick disease type C (NPC) maintenance and proliferation in hippocampus neurogenesis, and lipogenesis is required for neuronal differentiation [44]. These pathways show metabolic changes that occur during neural development. In (anaerobic) glucose metabolism, lipids, in addition to glucose, can be used as an additional energy source. In this sense, the importance of mitochondrial function in neural development is now acknowledged and garnering increased attention.

## 3. Mitochondrial and Neurodegenerative Disorder

A crucial phenomenon causing neurodegeneration is triggered by inadequate mitochondrial molecular control. When combined with additional dysfunctions, mitochondrial dysfunction can result in highly severe neuropathological conditions [45]. There is evidence that mitochondrial failure and respiratory-complex damage contribute to neuronal loss in neurodegenerative diseases like Parkinson’s and Alzheimer’s [46]. Mitochondrial respiratory chain complex I catalyses electron transport from NADH to the ubiquinone pool (Q), with proton exchange across the inner mitochondrial membrane [47]. This complex is one of the primary places where electrons are liberated and react with oxygen, producing ROS and causing oxidative stress [48]; it is made up of 44 structural subunits (seven of which are mtDNA encoded) and at least 14 ancillary/assembly components [49]. Clinical symptoms linked with complex I deficiency are diverse and not clearly defined. Furthermore, mitochondrial respiratory chain complex II is an essential component of both the Krebs cycle and the mitochondrial respiratory chain, both of which play critical roles in ATP generation [50]. Additional electrons from succinate are transported into the Q in complex II [51]. The formation of ROS by the mitochondrial complex II may have either beneficial or detrimental consequences depending on the physiological conditions. Increased mitochondrial ROS production has been associated with a number of pathophysiological disorders, including neurodegenerative diseases [52]. Complex II deficiency is uncommon, except in cases of mitochondrial dysfunction [53]. Biallelic mutations have been linked to congenital metabolic disorders, primarily affecting the central nervous system (CNS) or the heart (hypertrophic cardiomyopathy, leukodystrophy, Leigh syndrome, and encephalopathy) [54], whereas heterozygous mutations have been linked to cancer predisposition, specifically pheochromocytoma, and paraganglioma [55]. Mitochondrial respiratory chain complex III transports electrons from the Q to cytochrome c while pumping protons from the mitochondrial matrix into the intermembranous region [56]. Complex III insufficiency is a very frequent oxidative phosphorylation (OXPHOS) system impairment associated with a wide range of neurological diseases [57]. Complex III is made up of 11 structural subunits, two heme groups, and the iron–sulphur protein Rieske. Cardiomyopathy and encephalomyopathy have also been observed in clinical phenotypes with mutations in the mtDNA MTCYB gene [58]. The terminal complex of the electron transport chain, however, is mitochondrial respiratory chain complex IV or cytochrome c oxidase (COX). In a process linked to proton pumping across the inner mitochondrial membrane, it catalyses electron transfers from cytochrome c to reduce molecular oxygen (O_2_) and create two molecules of water. Complex IV deficits are caused by an unknown chemical mechanism [59]. The electrochemical gradient generates the synthesis of ATP by complex V in the OXPHOS system. The importance of the OXPHOS system stems from the fact that it is the primary energy source in most cell types and has a role in the control of the cell’s redox status and critical metabolic and signalling pathways ranging from pyrimidine synthesis to apoptosis regulation [60,61]. Mutations in structural COX subunits have been described, although the majority of abnormalities affect biogenesis proteins. Some proteins are strongly connected to specific elements of COX biogenesis [62], whereas others play a more varied role [63]. The majority of people with complex V deficiency developed it during their newborn period, resulting in multiorgan failure or severe brain injury and a high mortality rate [64].

Mutations in genes that encode components of the mitochondrial respiratory chain cause mitochondrial diseases with acute neurological symptoms (Figure 1) [65]. Defective mitochondrial enzyme activity, particularly pyruvate dehydrogenase and ketoglutarate dehydrogenase complexes, have been observed in Alzheimer’s disease patients. As previously stated, cytochrome oxidase (respiratory chain complex IV) is the most notably damaged mitochondrial enzyme and has been widely researched in AD [66]. The main phenomena of AD are: first, the abnormal deposition of amyloid plaques in the extracellular environment of the brain; and second, the dysfunction of the mitochondrial cascade is considered to be the second triggering phenomenon of this disease [67].

Many technologic and scientific advancements have recently been developed to provide new information on the putative processes of irreversible neurodegeneration, particularly the initial events leading to reversible neuronal cell death [68]. This will allow researchers to have a better understanding of neural susceptibility in mitochondrial disease [69]. Pluripotent cells differentiate into neurons and glial cells, allowing for the investigation of the nuclear genome and the mitochondrial genome to discover disease causes, therapeutic efficacy, and cell replacement therapies [70]. Furthermore, a number of nuclear mitochondrial genes found in specific groups of neurons and glial cells hold promise for understanding the disease’s specific processes, including the common tRNA point mutations MT-TL1 and MT-TK and also single large-scale mtDNA deletions [71,72].

## 4. Mitochondrial Dysfunction and Oxidative Stress

Oxidative stresses are an important factor in the pathogenesis of various neurological diseases. Mitochondria, being the centre of cellular metabolism, are a key regulatory system in redox balance and play a determining role in the development and progression of diseases [73]. Complexes I and III are assumed to be important sites for superoxide and other reactive oxygen species generation [74]. As a result, the highly reactive hydroxyl radical can damage macromolecules within mitochondria, such as lipids, proteins, and DNA [75], and unrepaired mitochondrial DNA damage can cause poor complex I and/or III function and produce a high production of superoxide [76,77]. In the synthesis of adenosine triphosphate, hydrogen ions derived from nicotinamide adenine dinucleotide and reduced flavin adenine dinucleotide in intermediary metabolism are transported along the complexes to molecular oxygen, resulting in the production of water [78], and protons are pumped across the mitochondrial inner membrane by complexes I, III, and IV [79]. The flow of these protons back into the mitochondrial matrix via complex V produces adenosine triphosphate [79]. Under typical physiological settings, 1–5% of the oxygen consumed is transformed to ROS [80]. As a result, most estimations indicate that mitochondria produce the majority of intracellular ROS. The generation of mitochondrial superoxide radicals occurs largely at two distinct locations in the electron-transport chain, namely complex I and complex III, with the latter being the primary site of ROS synthesis [81,82]. Any mitochondrial failure can result in an increase in reactive oxygen species (ROS), which disrupts mitochondrial calcium (Ca^2+^) homeostasis and causes depolarisation of the inner mitochondrial membrane potential. Such changes may result in free radical-induced cellular damage, such as membrane lipid peroxidation and DNA damage [83]. As a result, the activity of several major enzymatic antioxidants, including manganese superoxide dismutase (MnSOD), catalase, glutathione peroxidase/glutathione-S-transferase, and the thioredoxin and methionine sulphoxide reductase pathways, will be reduced, resulting in a redox imbalance linked to the central nervous system perturbation and neuron damage. Decreased levels of complex I proteins NDUFB8 (NADH ubiquinone oxidoreductase subunit B8) and NDUFS3 (NADH ubiquinone oxidoreductase subunit S3) increase reactive oxygen species (ROS) and inflammatory cytokines (cytokines), causing mitochondrial DNA damage and mutation, and high-level deletions of mtDNA, thus encoding protein subunits of mitochondrial complex I, III, IV, and V [84]. Thus, neuroinflammation is usually caused by the excessive activation of brain immune cells, specifically microglia and astrocytes, by DAMP molecules produced by injured and necrotic cells [84,85]. Microglia cells are in charge of removing damaged neurons as well as monitoring infections. Astrocytes, on the other hand, are responsible for the maintenance of brain structure and the regulation of synapses, as well as providing neuronal metabolic support [86]. Dysregulated activation of microglia and astrocytes leads to prolonged inflammasome activation, which leads to the establishment of low-grade chronic inflammation and, as a result, the development of age-related degenerative processes [87]. As a result, neuroinflammation promotes the release of cytokines and chemokines outside the brain [88] and, in rare situations, may result in blood-brain barrier rupture with peripheral immune cell invasion [89].

## 5. Mitochondria and Apoptosis Mechanism

The mitochondria are important organelles in regulating cell destiny because they may operate as “on-off” switches, governing autophagy and apoptosis [7]. The mitochondrial permeability transition pore is a multiprotein complex generated at the junction of the mitochondrial inner and outer membranes, where Bax, Bcl-2, and Bcl-XL are prevalent. The pore regulates matrix Ca^2+^ content, pH, transmembrane potential, and volume and works as a Ca^2+^-, voltage-, pH-, and redox-gated channel with a variety of permeability levels and very little ion selectivity. Members of the Bcl-2 family, which interact with the permeability transition pore complex, regulate the permeabilisation of mitochondrial membranes, which is a critical aspect of early cell death; this function can also be performed by the tau441 protein, α-synuclein, and β-amyloid oligomers, all of which are pathogenically linked to neurodegenerative diseases, owing to their binding to the proapoptotic BAK protein [90]. The regulation of membrane permeabilisation is associated with the regulation of mitochondrial bioenergetics and redox activities. The mitochondrial transition permeability pore is thus a potentially important element in neurodegenerative cell death (Figure 2). It has been demonstrated that the pore is implicated in oxidant-induced mitochondrial enlargement, Ca^2+^ release, and cell death [91]. Cellular apoptosis, on the other hand, is mediated by an intracellular signalling program involving a number of signalling molecules and cell organelles, most notably caspases, sphingomyelinases, Bcl-2-type proteins, and mitochondrial DNA cleavage [92]. This mechanism influences various conditions, including neurodegenerative diseases, such as Alzheimer’s disease (AD) and Parkinson’s disease.

Main consequences resulting from mitochondrial dysfunction include the alteration of mitochondrial homeostasis and bioenergetic failure, causing increased production of ROS, which can lead to neurodegenerative disorders and even induce cell death. 

## 6. Neuroprotective Activities of Bioactive Compounds

Bioactive substances are extra-nutritional elements found in foods, especially fruits, vegetables, and grains, that have the ability to alter metabolic processes while also offering health benefits. Secondary metabolites have been identified as having potential therapeutic qualities, such as hepatoprotective, hypoglycemic, antioxidant, anti-inflammatory, immunomodulatory, wound healing, cardioprotective, anti-cancer, and neuroprotective activities [93]. Their biological activities and nutritional values have made them the focus of numerous in vitro and in vivo studies in recent years [94]. Various plant products have been demonstrated to exhibit antioxidant activity, and plant-derived antioxidant vitamins, essential oils, flavonoids, and polyphenolic chemicals have been intensively studied as free radical scavengers and lipid peroxidation inhibitors [95]. Recent studies have shown that herbal extracts could potentially restore altered neurological capacities and antioxidant power due to the high presence of bioactive compounds such as carnosol, rosmarinic acid, oleanolic acid, and ursolic acid. It has been demonstrated that polyphenols, like curcumin, carnosol, and rosmarinic acid, reduce the expression levels of oxidised lipids and maintain the proper functioning of the different mitochondrial complexes, protecting the morphological aspects of the brain’s mitochondria and improving the biochemical functions of neurons [96]. The latter could even be considered as a good agent against the deleterious effect of heavy metal-induced brain injuries, like aluminium [97,98]. Because of their structural diversity, over 8000 phenolic compounds have previously been identified [99,100]. In general, phenolic chemicals are divided into two groups: flavonoids and non-flavonoids. Phenolic compounds have antioxidative properties due to their ability to scavenge free radicals and donate hydrogen atoms, electrons, and chelate metal cations; the antioxidant activity is closely related to their molecular structures, specifically the number and positions of the hydroxyl groups and the nature of the substitutions on the aromatic rings [101]. They are well-known for their ability to mediate neuroinflammation and neurodegenerative disorders by targeting toll-like receptor (TLR) pathways, particularly the TLR4 pathways. In this way, phenolic compounds that target TLR4 could be used as pharmacophores in the development of therapeutic treatments for neurological disorders [102]. Furthermore, the structure of flavonoids lends them a lipophilic property, allowing them to pass through the blood-brain barrier. As a result, interactions between cell membranes and flavonoids are critical because flavonoids embedded inside the lipid bilayer can cause changes in the lipid head and/or tails [103,104]. In this way, flavonoids are thought to be neuroprotective antioxidants because they limit the generation of free radicals by modulating cell signalling pathways involved in cell proliferation and survival, glutathione synthesis, and antioxidative protein expression [103]. Furthermore, they have been demonstrated to prevent ischemic-related apoptosis, amyloidogenic effects, and dopaminergic neuron loss by increasing neuron survival, tissue perfusion, and cerebral blood flow [105]. These effects are thought to be caused by flavonoids’ ability to bind to α-aminobutyric acid type A (GABAA) receptors in the central nervous system [106]. Phenolic acids have been shown to have neuroprotective effects in the central nervous system by alleviating ischemia, neuroinflammation, glutamate-induced toxicity, apoptosis, depression, memory impairment, and hearing and visual abnormalities. In recent years, there has been a significant deal of scientific interest in their potential involvement in protecting neurons and glial cells [107]. Polyphenols in the diet have been found to lower oxidative stress, which is involved in the initiation and progression of neurodegeneration. These substances have been shown to prevent neuronal death by lowering ROS levels, blocking caspase-3 activation, and enhancing redox activity [108].

Many in vivo studies on Wistar rats have strongly demonstrated that phytochemicals could repair impaired motor function and memory acquisition by modulating AChE activity, which is an enzyme that regulates ionic currents in excitable membranes and is necessary for nerve conduction at the neuromuscular junction as well as motor function [109].

The effects of some secondary compounds can play a very important role in reducing the effects of oxidative stress and regulating mitochondrial functions, among which we have:

Protective effect of quercetin:

Flavonoids have gained significant attention as antioxidants with intriguing pharmacological properties. Quercetin (3,3′,4′,5,6-pentahydroxyflavone) (Figure 3) is a polyphenolic system found in many fruits and vegetables, as well as grapes, citrus, and tea. Quercetin has been linked to a variety of health benefits, including anti-allergic, anti-rheumatic, anti-inflammatory, and antiviral properties. Furthermore, it inhibits ROS-producing enzymes and is known to protect neurons from oxidative stress-induced damage. It has also been observed that this substance can improve cognitive performance in patients with neurological diseases as well as cognitive function in neurobehavioral disorders [110]. A study on the impact of aluminium on the brain of rats showed that the administration of quercetin was effective in preventing aluminium exposure-induced cell death in the hippocampal region, which decreased the rate of ROS formation in the aluminium-intoxicated rat brain. As it was also shown in this work, quercetin prevents aluminium-induced cytochrome-c translocation, and upregulates Bcl-2, downregulates Bax, p53, and caspase activation, and reduces DNA damage [111].

Preventive role of Rutin against neural apoptosis and neuroinflammation:

Rutin (quercetin-3-rutinoside hydrate) (Figure 3), a well-known flavonoid found in a variety of foods and plants (including onion, oranges, apple, lemon, grapes, and tea), is known to have neuroprotective effects due to its antioxidative and anti-inflammatory properties [112]. Rutin’s anti-inflammatory effect may be explained by the inhibition of some important enzymes involved in inflammation and/or cell-signalled pathways, such as cyclooxygenase (COX) and lipoxygenase (LOX), protein kinase C (PKC), and phosphoinositide 3 kinases (PI 3 kinases), all of which contribute significantly to the production of inflammatory mediators like leukotrienes and prostaglandin. It can also suppress the expression of caspase-3, TNF, and NF-kB proteins, which are needed in order to facilitate access for inflammatory disorders [113]. Thus, those phytochemicals may be potential neuroprotective, neuromodulatory, and anti-inflammatory agents, and it is extremely apparent that their capacity to interact with protein and lipid kinase signalling cascades are mediated by these positive properties [114].

Through the production of ROS from mitochondria, oxidative stress is a common factor that plays a central role in the pathogenesis of several neurodegenerative diseases, and several reviews have indicated that phenolic compounds not only have potent antioxidant properties for free radical scavenging but they may also act on specific signalling regulatory pathways of inflammatory reactions [115,116].

Moreover, it was revealed that treatment with extracts containing antioxidant components offered effective protection against neuronal damage by decreasing the elevated TBARS concentration in brain mitochondrial and by increasing the activity of antioxidant enzymes superoxide dismutase (SOD), catalase, and glutathione peroxidase (GPx), which are critical antioxidant enzymes involved in cellular defence against the deleterious effects of ROS [117]. 

## 7. Conclusions

Because of its high metabolic activity and decreased cellular regeneration capability, oxidative stress is a crucial component at the beginning of neurodegenerative diseases. However, as molecular biology discoveries made the mitochondrial genome more accessible in recent years, interest in the mitochondrion’s role in disease aetiology has exploded. To begin, certain rare inherited disorders have been linked to abnormalities in the mitochondrial DNA. Other researchers have lately suggested that mitochondria may play a role in ageing by generating tissue-damaging reactive oxygen molecules or by impairing and depriving the cell of the energy it requires to function [118]. The revelation that mitochondria play a critical part in the regulation of programmed cell death, or apoptosis, is one of the most significant recent advances. We know that mitochondria play an important part in cell life and death decisions; mitochondria can cause cell death in a variety of ways: by interfering with their own electron transport and energy metabolism, initiating the mitochondrial permeability transition, and releasing and/or activating apoptosis-mediating proteins. All of these pathways may assist in explaining how mitochondrial abnormalities lead to neuronal death or malfunction in ischemia/reperfusion injury, as well as in human degenerative disorders such as Alzheimer’s disease, Parkinson’s disease, amyotrophic lateral sclerosis, and Huntington’s disease [119]. Furthermore, the use of bioactive substances in mitotherapy will significantly reduce the negative impact of ROS production and cell viability [120,121]. Some research is focusing on the development of treatment techniques to offset the effects of ROS and RNS. While the human body has a natural antioxidant defence mechanism that seeks to prevent free radicals from reacting with biological components, exogenous injection of antioxidative substances is critical. Several dietary antioxidants have been studied for their neuroprotective effects in the treatment of neurodegenerative diseases in this manner. All of these studies are being conducted in order to identify the most important secondary compounds that can aid in the development of a new family of mitochondrially targeted pharmaceuticals capable of remedying neurodegenerative disease states and enabling longevity and healthy ageing for the vast majority of people.

## Figures and Tables

**Figure 1 cells-11-00257-f001:**
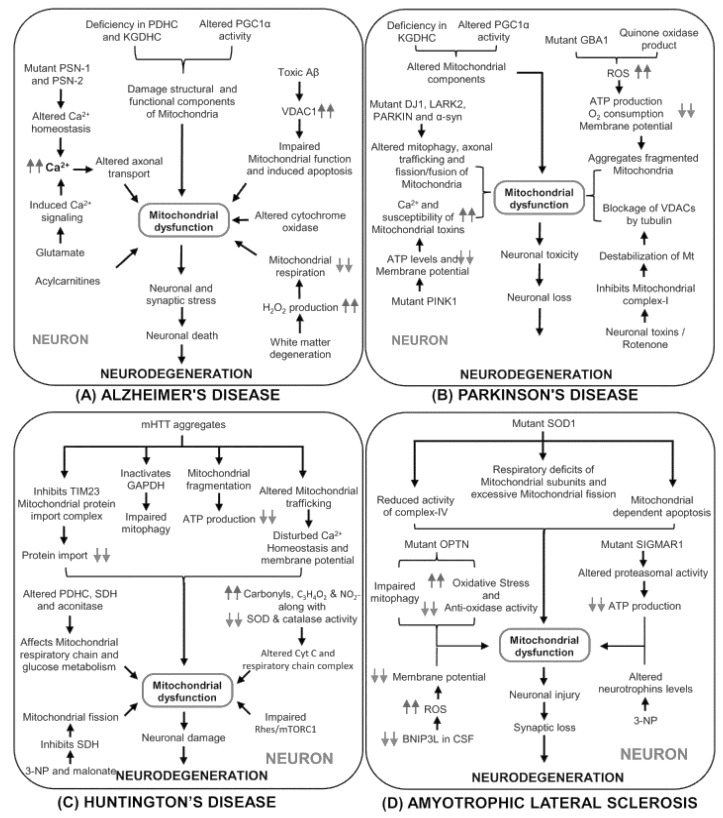
The molecular process by which mitochondrial failure mediates the pathogenesis of neurodegenerative diseases. PGC1, peroxisome proliferator-activated receptor gamma coactivator 1-alpha; PSN-1 and PSN-2, presenilin-1 and presenilin-2; VDAC1, voltage-dependent anion channel 1; GBA1, glucocerebrosidase 1; GAPDH, glyceraldehyde-3-phosphate dehydrogenase; PINK1, PTEN-induced putative kinase 1; 3-NP, 3-nitropropionic Acid; SOD1, superoxide dismutase; OPTN, optineurin; SIGMAR1, sigma-1 receptor (Sig-1R); BNIP3L, pro-apoptotic mitochondrial proteins; CSF, cerebrospinal fluid. (**A**) molecular mechanisms linked to mitochondrial function and the onset of Alzheimer’s disease. (**B**) Parkinson’s disease development and altered mitochondrial components (**C**) Changes in the mitochondrial respiration chain and the beginning of Huntington’s disease. (**D**) The impact of oxidative stress in mitochondrial failure and neuronal dysfunction in amyotrophic lateral sclerosis.

**Figure 2 cells-11-00257-f002:**
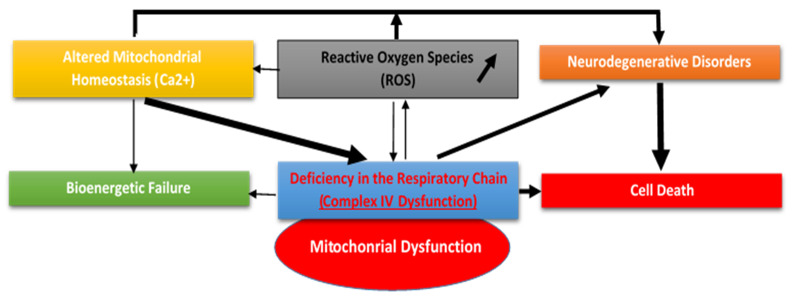
Neurodegenerative disorders related to mitochondrial dysfunction.

**Figure 3 cells-11-00257-f003:**
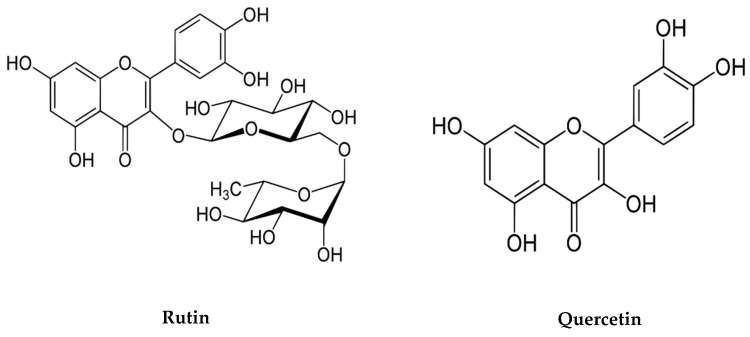
Chemical structure of Quercetin and Rutin.

## Data Availability

Not applicable.

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
