# Peer review of "Role of Bioactive Compounds in the Regulation of Mitochondrial Dysfunctions in Brain and Age-Related Neurodegenerative Diseases"

_cells, 2022, doi:10.3390/cells11020257_

Round 1
Reviewer 1 Report
The authors describe the potential neuroprotective role of bioactive compounds in mitochondrial function and bioenergetics.
English writing of the manuscript needs great improvements.
In the title of the manuscript “Role of bioactive compounds in the regulation of mitochondrial functions in brain and age-related neurodegenerative diseases” do authors refer to healthy brain. It is not clear as it is.
The authors should rephrase some sentences, e.g.:
Line 42 – “Molecular mechanisms involved in all cells, and mitochondria energy production occurs in all aerobically respiring mammalian cells”
Line 45. “….generate ATP in the most efficient way possible via oxidative phosphorylation (OXPHOS)…”
Line 49 – “…this ability to adapt structurally and functionally is made up of a variety of cellular and molecular mechanisms.” Made up should be changed
Line 50 – “With the excitatory neurotransmitter glutamate [3, 4], The most extensively researched members of these three kinds of signals are brain derived neurotrophic factor [5] and neural cell adhesion molecule [6]. Besides from cell signaling pathways and their interest in function regulation and energy production.” These sentences do not make sense.
Line 58- what do authors mean with calcium transmission? Should It be homeostasis
Line 59 – “Mitochondria can be found along the length of axons and at presynaptic terminals.” Can authors clarify this sentence, no mitochondria are found at postsynaptic terminals?
Line 76 – “prevent or eliminate oxidative stress-related disorders by counteracting the detrimental effects of ROS/RNS”. Instead of disorders should be effects since neurodegenerative disorders have multifactorial causes.
Line 82 – “Furthermore, the absence and/or presence imbalances of ROS/RNS complicates the identification of appropriate antioxidant drugs.”
Line 89 – “particularly at synapses where the energy demand is the greatest”. High instead of is the greatest
Line 90 – “Mitochondria are intracellular organelles found in all nucleated cells in many copies and they have their unique genome”.
Line 92 –“The perturbation of the function and essentially by in the CNS, mitochondrial ROS production is linked in both …”
Line 95- “Another major consequence of Ca2+ transport and mitochondrial ROS production in diverse neuronal preparations is mitochondrial interactions with neuronal activity under physiological conditions [25, 26].” What do authors mean with Ca2+ transport and neuronal preparations.
Line 98 – It is not clear why the authors start referring glutamate release in this section. Seems out of context with the rest of the text.
Line 103 – “Following the discovery that mitochondria are a major generator of cellular ROS under physiological conditions, electrons transferred in mitochondria do not follow the normal electron transfer order and interact with oxygen to form superoxide or hydrogen peroxide [30].” This sentence is confusing
Line 118 – The authors use the abbreviation [Ca2+]c without defining it
Line 122 – “Among the most important functions of mitochondria are cell growth, reproduction, neurogenesis and differentiation, efficient neurotransmission, and to maintain cell membranes ionic gradients”. The authors should rephrase since mitochondria regulate key factors in the activation of cell signaling pathways involved in these processes.
Line 131- “The accumulation of mitochondrial biogenesis in conjunction with ATP enabled differentiation to support basic cellular activities in neurite formation”. This sentence is not clear.
Line 140 – “Transcriptional gene regulation is an example of a process that governs neurogenesis [42]. During neural development, transcription factors affect the transcriptome profile of cells.” The 2 sentences have the same meaning.
Line 145 – NPC is not defined
Line 152 – “Neurodegeneration is caused by improper mitochondrial molecular control.” The authors should rephrase this sentence since other factors are involved
Line 155 – “neurodegenerative diseases like aging”. Aging is not a neurodegenerative disease.
Line 157 –“ as well as vector illustration proton pumping across the inner mitochondrial membrane”. What do authors mean with this sentence?
Line 166 - “Depending on the physiological circumstances, the generation or activation of ROS by the mitochondrial complex II may have either harmful or beneficial effects”. This sentence is not clear
Line 184- “it catalyses electron transfers from cytochrome c to decrease molecular oxygen (O2) and create two molecules of water”.
Shouldn’t it be reduce instead of decrease?
Line 186 – “The fifth enzyme of the OXPHOS system, mitochondrial respiratory chain complex V, is found in the mitochondrial inner membrane [60]. Mutations in structural COX subunits have been described…”. The COX subunits are from Cx IV.
Line 193- “Mutations in genes encoding components of the mitochondrial respiratory chain, on the other hand, result in mitochondrial diseases marked by neurological symptoms (Fig.1).” Mitochondrial dysfunction are early features of these diseases which are not mitochondrial diseases by definition.
Line 211 – Can the authors clarify “Many technologies”
Line 217 – “Furthermore, a number of nuclear mitochondrial genes found in specific groups of neurons and glial cells hold promise for understanding the disease's specific processes. [70, 71]”. Can the authors specify which genes.
Line 227 – “and unrepaired mitochondrial DNA damage that results in deficient complex I and/or III function can result in increased electron reduction of O2 to form superoxide”
line 239 “Any mitochondrial defect can result in an increase in reactive oxygen species (ROS), which prevents appropriate mitochondrial calcium (Ca2+) levels and produces depolarization of the inner mitochondrial membrane potential”. What do authors mean with prevents appropriate mitochondrial calcium levels?
Line 247 “perturbation and neuroin It was determined that Complex I”
Line 251 – The authors refer neuroinflammation as a consequence of ROS and mitochondria dysfunction but don't explain the mechanism.
Line 283 “dysfunction, including Alteration of mitochondrial homeostasis and bioenergetic failure causing increased production of...”
There is no reference to fig. 2 in the text.
Line 299/ 300 – “it has been demonstrating …..the expression levels oxidised lipids”
Line 334 -338 the sentence is too long.
Line 379 - TBARS is not defined
Line 384 – “Because of its high metabolic activity…” please define its
The authors only refer the benefits of Quercetin and Rutin ignoring other polyphenolic compounds such as Resveratrol or Curcumin that already have shown to improve mitochondrial dysfunction and ameliorate oxidative stress.
Author Response
Dear Colleague
We appreciate your very pertinent remark, which assisted us in improving the content of this manuscript.
You have been informed that almost all of your comments and suggestions have been considered.
Reviewer 2 Report
The review seems to be interesting and relevant, although it is not without some drawbacks. In particular, in Section 5, the authors talk about the multiprotein structure of the pore, but only mention proteins of the BCL-2 family (not invоlving in mPT pore of inner mito membrane). It would be possible to reflect this question more fully using information in other articles (for example - DOI: 10.1002 / med.21715, DOI: 10.1038 / s41580-021-00433-y, DOI: 10.1111 / febs.16254)
Section 6 would be well formatted to make the subsections more obvious and the table with the formulas of the compounds mentioned is clearly needed.
Line 136 - You need to replace the term decoupling with uncoupling.
Edit lines 50-53.
Author Response
Dear Colleague
We appreciate your very pertinent remark, which assisted us in improving the content of this manuscript.
Best regard
Round 2
Reviewer 1 Report
The quality of the manuscript has improved considerably.
Just have 2 remarks:
Line 168 - Niemann-Pick disease type C (NPC)
should it be Neural progenitor cells?
line 212 - The fifth enzyme of the OXPHOS system, mitochondrial respiratory chain complex IV, is found in the mitochondrial inner membrane.
Author Response
Dear colleague,
Line 168 - Niemann-Pick disease type C (NPC)
Yes, NPC indicate Niemann-Pick disease type C, however it has been reported that NPC 1 gene deficiency could lead to lack of the self-renewal ability of neural stem cells in Niemann pick type C disease.
line 212 - The fifth enzyme of the OXPHOS system, mitochondrial respiratory chain complex IV, is found in the mitochondrial inner membrane.
we replace this sentence by :
The electrochemical gradient generates the synthesis of ATP by complex V in the OXPHOS system. Because the importance of the OXPHOS system stems from the fact that it is the primary energy source in most cell types, as well as its role in the control of the cell's redox status, as well as critical metabolic and signaling pathways ranging from pyrimidine synthesis to apoptosis regulation.
Thank you infinitely for your relevant remarks.
Best regards